# TarGF: Learning Target Gradient Field to Rearrange Objects without Explicit Goal Specification

**Mingdong Wu**[* 1, 3], **Fangwei Zhong**[* 2, 3], **Yulong Xia**[1], **Hao Dong**[1, 4]
[1] Center on Frontiers of Computing Studies, School of Computer Science, Peking University
[2] School of Intelligence Science and Technology, Peking University
[3] Beijing Institute for General Artificial Intelligence (BIGAI)
[4] Peng Cheng Laboratory
{wmingd, zfw, hao.dong}@pku.edu.cn

## Abstract

Object Rearrangement is to move objects from an initial state to a goal state. Here, we focus on a more practical setting in object rearrangement, *i.e.*, rearranging objects from shuffled layouts to a normative target distribution without explicit goal specification. However, it remains challenging for AI agents, as it is hard to describe the target distribution (goal specification) for reward engineering or collect expert trajectories as demonstrations. Hence, it is infeasible to directly employ reinforcement learning or imitation learning algorithms to address the task. This paper aims to search for a policy only with a set of examples from a target distribution instead of a handcrafted reward function. We employ the score-matching objective to train a ***Tar**get **G**radient **F**ield (TarGF)*, indicating a direction on each object to increase the likelihood of the target distribution. For object rearrangement, the TarGF can be used in two ways: 1) For model-based planning, we can cast the target gradient into a reference control and output actions with a distributed path planner; 2) For model-free reinforcement learning, the TarGF is not only used for estimating the likelihood-change as a reward but also provides suggested actions in residual policy learning. Experimental results in ball rearrangement and room rearrangement demonstrate that our method significantly outperforms the state-of-the-art methods in the quality of the terminal state, the efficiency of the control process, and scalability. The code and demo videos are on https://sites.google.com/view/targf.

## 1 Introduction

As shown in Fig. 1, we consider object rearrangement *without explicit goal specification* [1, 2], where the agent is required to manipulate a set of objects from an initial layout to a normative distribution. This task is taken for granted by humans [3] and widely exists in our daily life, such as tidying up a table [4], placing furniture [5], and sorting parcels [6].

However, compared with conventional robot tasks [7, 8, 9, 10, 11], there are two critical challenges in our setting: 1) Hard to clearly define the target state (goal), as the target distribution is diverse and the patterns are difficult to describe in the program, *e.g.*, how to evaluate the *tidiness*

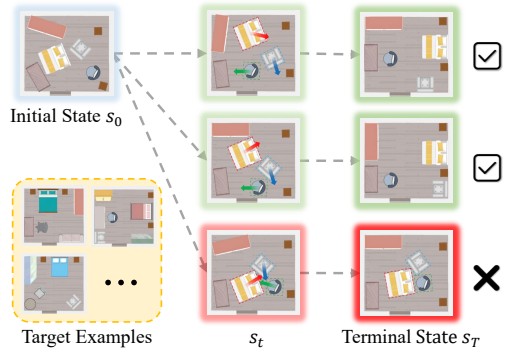

Figure 1: Our goal: Learning to rearrange objects **without explicit goal specification**.

---

* indicates equal contribution

36th Conference on Neural Information Processing Systems (NeurIPS 2022).

quantitatively? As a result, it is difficult to manually design a reward/objective function, which is essential to modern control methods, *e.g.*, deep reinforcement learning (DRL) [12]. 2) In the physical world, the agents also need to find an executable path to reach the target state efficiently and safely, *i.e.*, finding a short yet collision-free path to reach the target distribution. To this end, the agents are required to jointly learn to evaluate the quality of the state (reward) and move objects to increase the quality efficiently. In other words, the agents should answer the two questions: *What kind of layouts does the user prefer*, and *How to manipulate objects to meet the target distribution*?

Thus, searching for a policy without dependence on the explicit goal state and shaped reward is necessary. Imitation learning (IL) can learn a reward function [13, 14, 15] or learn a policy [16, 17, 18, 19] from the expert's supervision, *e.g.*, demonstrating expected trajectories. However, collecting a large number of expert trajectories is expensive. Recently, example-based reinforcement learning (RL) [20, 21] has tried to reduce the reliance on expert trajectories, *i.e.*, learning a policy only with examples of successful outcomes. However, it is still difficult to learn a classifier to provide accurate reward signals for learning in the case of high-dimensional state and action space, as there are many objects to control in object rearrangements.

In this work, we separate the task into two stages: 1) learning a score function to estimate the target-likelihood-change between adjacent states, and 2) learning/building a policy to rearrange objects with the score function. Inspired by the recent advances in score-based generative models [22, 23, 24, 25, 26], we employ the denoising score-matching objective [27] to train a *target score network* for estimating the *target gradient field*, *i.e.*, the gradient of the log density of the target distribution. The target gradient field can be viewed as object-wise guidance that provides a pseudo velocity for each object, pointing to regions with a higher density of the target distribution, *e.g.*, tidier configurations. To rearrange objects in the physical world, we demonstrate two ways to leverage the target gradient field in control. 1) We integrate the gradient-based guidance with a model-based path planner to rearrange objects without collision. 2) We further derive a reward function for reinforcement learning by estimating the likelihood change via the target gradient. With the TarGF-based reward and a TarGF-guided residual policy network [28], we can train an effective RL-based policy.

In experiments, we evaluate the effectiveness of our methods on two typical rearrangement tasks: *Ball Rearrangement* and *Room Rearrangement*. First, we introduce three metrics to analyse the quality of the terminal state and the efficiency of the control process. In *Ball Rearrangement*, we demonstrate that our control methods can efficiently reach a high-quality and diverse terminal configuration. We also observe that our methods can find a path with fewer collisions and shorter lengths to converge compared with learning-based baselines [14, 21] and planning-based baselines [29]. In *Room Rearrangement*, we further evaluate our methods' feasibility in complex scenes with multiple heterogeneous objects.

Our contributions are summarised as follows:

- We introduce a novel score-based framework that estimates a *target gradient field (TarGF)* via denoising score matching for object rearrangement.

- We point out two usages of the target gradient field in object rearrangement: 1) providing guidance in moving direction; 2) deriving a reward for learning.

- We conduct experiments to demonstrate that both traditional planning and RL methods can benefit from the target gradient field and significantly outperform the state-of-the-art methods in sample diversity, realism and safety.

## 2 Related Works

### 2.1 Object and Scene (Re-)Arrangement

The object (re-)arrangement is a long-studied problem in robotics community [2, 30, 31] and the graphics community [32, 33, 34, 35, 36]. However, most of them focus on manually designing rules/ energy functions to find a goal [2, 30, 31] or synthesising a scene configuration [33, 34, 35, 36] that satisfies the preference of the user. The most recent work [4] tries to learn a GNN to output an arrangement tailored to human preferences. However, they neglect the physical process of rearrangement. Hence, the accessibility and transition costs from the initial and target states are not guaranteed. Considering the physical interaction, the recent works on scene rearrangement [37, 38] mitigate the goal specification problem by providing *a goal scene* to the agent, making the reward shaping feasible. Concurrent works [39, 40] also notice the necessity of automatic goal inference for

tying rooms and exploit the commonsense knowledge from Large Language Model (LLM) or memex graph to infer rearrangements goals when the goal is unspecified. Another concurrent work [41] aims at discovering generalisable spatial goal representations via graph-based active reward learning for 2D object rearrangement. In this paper, we focus on a more practical setting of rearrangement: *how to estimate the similarity (i.e., the target likelihood) between the current state and the example sets and manipulate objects to maximise it*. Here, only a set of positive target examples are required to learn the gradient field rather than prior knowledge about specific scenarios.

## 2.2 Learning without Reward Engineering

Without hand-engineered reward, previous works explore algorithms to learn a control policy from expert demonstrations via behavioural cloning [16, 17, 18, 13] or inverse reinforcement learning [42, 15, 43, 44]. Imitation learning (IL) [16, 17, 18] aims to directly learn a policy by cloning the behaviour from the expert trajectories. The inverse reinforcement learning (IRL) [42, 15, 43, 44] tries to learn a reward function from data for subsequent reinforcement learning. Considering the difficulty in collecting expert trajectories, some example-based methods are proposed to learn a policy only with a set of successful examples [20, 14, 21]. The most recent work is RCE [21] which directly learns a recursive classifier from successful outcomes and interactions for policy optimisation. We also try to develop an example-based control method for rearrangement with a set of examples. Our method can learn the target gradient field from the examples without any additional effort in reward engineering. The target gradient field can provide meaningful reward signals and action guidance for reinforcement learning and traditional path planners. To the best of our knowledge, such usage of gradient field is not explored in previous works.

## 3  Problem Statement

We aim to learn a policy that moves objects to reach states close to a target distribution without relying on reward engineering. Similar to the *example-based control* [21], the grounded *target distribution* $p_{tar}(\mathbf{s})$ is unknown to the agent, and the agent is only given a set of *target examples* $S^* = \{\mathbf{s}^*\}$ where $\mathbf{s}^* \sim p_{tar}(\mathbf{s})$. In practice, these examples can be provided by the user without accessing the dynamics of the robot. The agent starts from an initial state $\mathbf{s}_0 \sim p_0(\mathbf{s}_0)$, where $N$ objects are randomly placed in the environment. At time step $t$, the agent takes action $\pi(\mathbf{a}_t|\mathbf{s}_t)$ imposed on the objects to reach the next state with dynamics $p(\mathbf{s}_{t+1}|\mathbf{s}_t, \mathbf{a}_t)$. Here, the goal of the agent is to search for a policy $\pi^*(\mathbf{a}_t|\mathbf{s}_t)$ that maximises the *discounted log-target-likelihood* of the future states:

$$\pi^* = \arg\max_{\pi} E_{\rho(\mathbf{s}_0), \tau \sim \pi} \left[ \sum_{\mathbf{s}_t \in \tau} \gamma^t \log p_{tar}(\mathbf{s}_t) \right] \tag{1}$$

where $\gamma \in (0, 1]$ denotes the discount factor. Notably, we assume that $p_{tar}(\mathbf{s}) > 0$ everywhere since we can perturb the original data with a tiny noise (*e.g.*, $\mathcal{N}(0, 0.00001)$) to ensure the perturbed density is always positive. [22] also used this trick to tackle the manifold hypothesis issue. This objective reveals three challenges: **1) Inaccessibility problem:** The grounded target distribution $p_{tar}$ in Eq. 1 is inaccessible. Thus, we need to learn a function to approximate the *log-target-likelihood* for policy search. **2) Sparsity problem:** The log-target-likelihood is sparse in the state space due to the *Manifold Hypothesis* mentioned in [22]. Hence, even if we have access to the target distribution $p_{tar}$, it is still difficult to explore the high-density region. **3) Adaptation problem:** The dynamics $p(\mathbf{s}_{t+1}|\mathbf{s}_t, \mathbf{a}_t)$ is also inaccessible. To maximise the cumulative sum in Eq. 1, the agent is required to adapt to the dynamics to efficiently increase the target likelihood.

To address these problems, we partition the task into 1) learning to estimate the log-target-likelihood (reward) of the state and 2) learning/building a policy to rearrange objects to adapt to the dynamics.

## 4  Method

### 4.1  Motivation

Estimating the *gradient* of the log-target-likelihood $\nabla_{\mathbf{s}} \log p_{tar}(\mathbf{s})$ can tackle the first two problems mentioned in Sec. 3: For the *inaccessibility* problem, we can approximate the likelihood increment

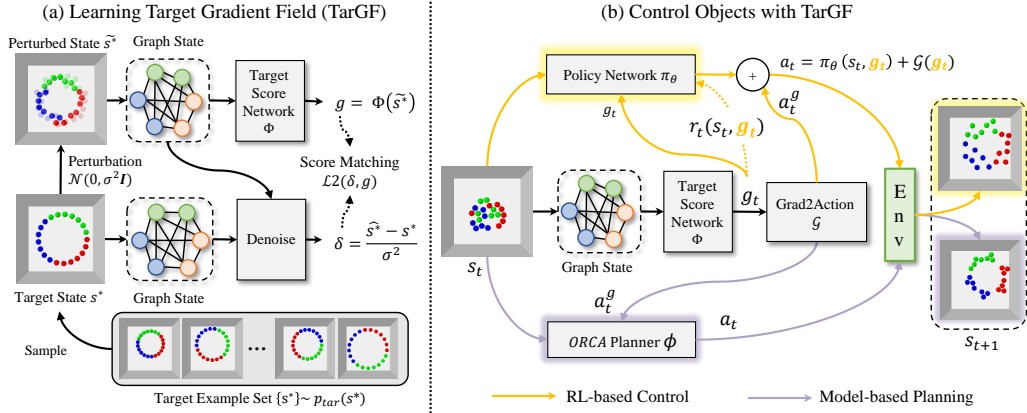

Figure 2: An overview of our method. (a) We train the target score network via score matching. The target examples are first perturbed by Gaussian noise. The target score network is forced to match the denoising direction of the perturbation. (b) Our framework is based on the trained target score network. The target score network provides exploration guidance and reward estimation for RL in the model-free setting. In the model-based setting, the TarGF provides reference velocities for a model-based planner based on ORCA. The planner then outputs collision-free velocities for objects.

between two adjacent states by the first-order Taylor expansion $\log p_{tar}(\mathbf{s}_{t+1}) - \log p_{tar}(\mathbf{s}_t) \approx \langle \nabla_{\mathbf{s}} \log p_{tar}(\mathbf{s}_t), \mathbf{s}_{t+1} - \mathbf{s}_t \rangle$. This helps to derive a surrogate objective (*i.e.*, Eq. 4) of Eq. 1.

For the *sparsity* problem, we can leverage the gradient field to help with exploration since the gradient $\nabla_{\mathbf{s}} \log p_{tar}(\mathbf{s})$ indicates the fastest direction to increase the target likelihood. To estimate $\nabla_{\mathbf{s}} \log p_{tar}(\mathbf{s})$, we employ the score-matching [23], a novel generative model that achieved impressive results in many research areas recently[24, 26, 25], to train a *target score network* $\boldsymbol{\Phi}_{tar} : \mathcal{S} \to \mathcal{S}$.

To address the *adaptation* problem, we further demonstrate two approaches to incorporate the trained target gradient field $\boldsymbol{\Phi}_{tar}$ with control algorithms for object rearrangement: 1) For the model-based setting, our framework casts the target gradient into a reference control and outputs an action with a distributed path planner. 2) For the model-free setting (reinforcement learning), we leverage the target gradient to estimate reward and provide suggested action in residual policy learning [28].

In the following, we will first introduce how to train the target gradient field $\boldsymbol{\Phi}_{tar}$ via score-matching in Sec. 4.2 and then introduce our rearrangement framework under the model-based and model-free settings in Sec. 4.3 and Sec. 4.4 respectively.

## 4.2 Learning the Target Gradient Field from Examples

The target score network $\boldsymbol{\Phi}_{tar}$ is the **core module** of our framework, which aims at estimating the *score function* (*i.e.*the gradient of log-density) of the target distribution $\nabla_{\mathbf{s}} \log p_{tar}(\mathbf{s})$.

To train the target score network, we adopt the Denoising Score-Matching (DSM) objective proposed by [27], which can guarantee a reasonable estimation of the $\nabla_{\mathbf{s}} \log p_{tar}(\mathbf{s})$. In training, we pre-specify a noise distribution $q_\sigma(\widetilde{\mathbf{s}}|\mathbf{s}) = \mathcal{N}(\widetilde{\mathbf{s}}; \mathbf{s}, \sigma^2 I)$. Then, DSM matches the output of the target score network with the score of the perturbed target distribution $q_\sigma(\widetilde{\mathbf{s}}) = \int q_\sigma(\widetilde{\mathbf{s}}|\mathbf{s})p_{tar}(\mathbf{s})d\mathbf{s}$:

$$\frac{1}{2}\mathbb{E}_{q_\sigma(\widetilde{\mathbf{s}}|\mathbf{s}),p_{tar}(\mathbf{s})}\left[||\boldsymbol{\Phi}_{tar}(\widetilde{\mathbf{s}}) - \nabla_{\widetilde{\mathbf{s}}} \log q_\sigma(\widetilde{\mathbf{s}}|\mathbf{s})||_2^2\right] = \frac{1}{2}\mathbb{E}_{q_\sigma(\widetilde{\mathbf{s}}|\mathbf{s}),p_{tar}(\mathbf{s})}\left[||\boldsymbol{\Phi}_{tar}(\widetilde{\mathbf{s}}) - \frac{\mathbf{s}-\widetilde{\mathbf{s}}}{\sigma^2}||_2^2\right] \quad (2)$$

The DSM objective guarantees the optimal score network satisfies $\boldsymbol{\Phi}^*_{tar}(\mathbf{s}) = \nabla_{\mathbf{s}}q_\sigma(\mathbf{s})$ almost surely. When $\sigma$ is small enough, we have $\nabla_{\mathbf{s}}q_\sigma(\mathbf{s}) \approx \nabla_{\mathbf{s}} \log p_{tar}(\mathbf{s})$, so that $\boldsymbol{\Phi}^*_{tar}(\mathbf{s}) \approx \nabla_{\mathbf{s}} \log p_{tar}(\mathbf{s})$.

In practice, we adopt a variant of DSM proposed by [23], which conducts DSM under different noise scales simultaneously. This way, we can efficiently try different noise scales after only one training. The target score network is implemented as a graph-based network $\boldsymbol{\Phi}_{tar}(\mathbf{s}, t)$, which is conditioned on a noise level $t \in (0, 1]$. The input state $\mathbf{s} = [\mathbf{s}^1, \mathbf{s}^2, ..., \mathbf{s}^N]$ is constructed as a fully-connected graph that takes each object's state (*e.g.*position, orientation) $\mathbf{s}^i \in \mathbb{R}^{f_s}$ and static properties (*e.g.*category, bounding-box) $\mathbf{p}^i \in \mathbb{R}^{f_p}$ as the node feature $[\mathbf{s}^i, \mathbf{p}^i] \in \mathbb{R}^{f_s+f_p}$. The input graph is pre-processed

by linear feature extraction layers and then passed through several Edge-Convolution layers. After message-passing, the output on the i-th node serves as the component of the target gradient on the i-th object's state $\boldsymbol{\Phi}_{tar}^i(\mathbf{s}) \in \mathbb{R}^{f_s}$. We defer structure details to Appendix. B.

## 4.3 Model-based Planning with the Target Gradient Field

The target gradient $\mathbf{g} = \boldsymbol{\Phi}_{tar}(\mathbf{s})$ can be translated into a *gradient-based action* $\mathbf{a}^g = \mathcal{G}(\mathbf{g})$, where $\mathcal{G}$ denotes the gradient-to-action translation. A gradient over the state space can be viewed as velocities imposed on each object. For instance, if the state of each object is a 2-dimensional position $\mathbf{s}_i = [x, y]$, then the target gradient component on the i-th object can be viewed as linear velocities on two axes $\boldsymbol{\Phi}_{tar}(\mathbf{s})^i = [v_x, v_y]$. When the action space is also velocities, we can construct $\mathbf{a}^g$ by simply projecting the target gradient into the action space $\mathbf{a}^g = \mathcal{G}(\mathbf{g}) = \frac{v_{max}}{||\mathbf{g}||_2} \cdot \mathbf{g}$, where the $v_{max}$ denotes the speed limit. Following the gradient-based action, objects will move towards directions that may increase the likelihood of the next state, which meets the need for the rearrangement task.

However, the gradient-based action will potentially lead objects to collide with each other since the target gradient cannot infer the environment constraints from only target examples.

To adapt the gradient-based action to the environment dynamics, we incorporate the target gradient field with a model-based off-the-shelf collision-avoidance algorithm *ORCA*. The ORCA planner $\phi$ takes the gradient-based action $\mathbf{a}^g$ as reference velocities and then outputs the collision-free velocities $\mathbf{a} = \phi(\mathbf{a}^g)$ with the least modification to the reference velocities. In this way, the adapted action can lead objects to move towards a higher likelihood region efficiently and safely. For more details of the ORCA planner, we defer to Appendix B.2.

## 4.4 Learning Policy with the Target Gradient Field

The model-based approach to 'correct' the target gradient assumes the objects are of circular shape. This limits the scope of this method when objects are of non-circular shape (*e.g.*, furniture). To this end, we propose a *model-free approach* based on reinforcement learning (RL) for object rearrangement, where the agent needs to adapt the dynamics via online interactions.

To tackle the inaccessibility problem mentioned in Sec. 3, we first derive an equivalent $J^*(\pi)$ of the original objective in Eq. 1 by subtracting a constant from the original objective:

$$
J(\pi) \iff \mathbb{E}_{\rho(\mathbf{s}_0), \tau \sim \pi}\left[\sum_{\mathbf{s}_t \in \tau} \gamma^t \log p_{tar}(\mathbf{s}_t)\right] - \underbrace{\frac{\mathbb{E}_{\rho(\mathbf{s}_0)}[\log p_{tar}(\mathbf{s}_0)]}{1 - \gamma}}_{constant \quad C}
$$
$$
= \mathbb{E}_{\rho(\mathbf{s}_0), \tau \sim \pi}\left[\sum_{1 \le t \le T} \gamma^t \sum_{1 \le k \le t} [\log p_{tar}(\mathbf{s}_k) - \log p_{tar}(\mathbf{s}_{k-1})]\right] \overset{def}{=} J^*(\pi)
$$
(3)

Further, we derive a surrogate objective $\hat{J}(\pi)$ to approximate the $J^*(\pi)$. We notice that in most physical simulated tasks, the distance between two adjacent states $||\mathbf{s}_t - \mathbf{s}_{t-1}||_2$ is quite small, which inspires us to approximate the log-likelihood-change of the adjacent states using the Taylor expansion:

$$
J^*(\pi) \approx \mathbb{E}_{\rho(\mathbf{s}_0), \tau \sim \pi}\left[\sum_{1 \le t \le T} \gamma^t \underbrace{\sum_{1 \le k \le t} \langle \boldsymbol{\Phi}_{tar}(\mathbf{s}_{k-1}), \mathbf{s}_k - \mathbf{s}_{k-1} \rangle}_{r_t}\right] \overset{def}{=} \hat{J}(\pi)
$$
(4)

In this way, we can optimise the surrogate objective $\hat{J}(\pi)$ by assigning $\sum_{1 \le k \le t} \langle \boldsymbol{\Phi}_{tar}(\mathbf{s}_t), \mathbf{s}_k - \mathbf{s}_{k-1} \rangle$ as the immediate reward. In practice, we simply assign the last term of the cumulative summation as the immediate reward $r_t = \langle \boldsymbol{\Phi}_{tar}(\mathbf{s}_t), \mathbf{s}_t - \mathbf{s}_{t-1} \rangle$, and this version is shown to be the most effective.

To further tackle the sparsity problem, our idea is to build a residual policy $\pi_\theta$ upon the gradient-based action $\mathbf{a}_t^g$ mentioned in Sec. 4.3, as $\mathbf{a}_t^g$ helps to explore the regions with a high target likelihood:

$$
\underbrace{\mathbf{a}_t = \pi_\theta(\mathbf{s}_t, \mathbf{g}_t) + \mathbf{a}_t^g}_{output\,action}, \quad \underbrace{\mathbf{a}_t^g = \mathcal{G}(\mathbf{g}) = \frac{v_{max}}{||\mathbf{g}_t||_2} \cdot \mathbf{g}_t}_{gradient-based\,action}, \quad \underbrace{\mathbf{g}_t = \boldsymbol{\Phi}_{tar}(\mathbf{s}_t)}_{target\,gradient}
$$
(5)

This approach is similar to the residual policy learning proposed by [28]. The $\mathbf{a}_t^g$ serves as an initial policy, and the residual policy network $\pi_\theta$ takes the target gradient $\mathbf{g}_t$ and the current state $\mathbf{s}_t$ as input and outputs $\pi(\mathbf{s}_t, \mathbf{g}_t)$ to 'correct' the $\mathbf{a}_t^g$. The agent finally outputs a 'corrected' action $\mathbf{a}_t = \pi(\mathbf{s}_t, \mathbf{g}_t) + \mathbf{a}_t^g$. In this way, the agent can benefit from the efficient exploration and have less training burden as it 'stands on the giant's shoulder' (*i.e.*the gradient-based action $\mathbf{a}_t^g$).

We employ the multi-agent Soft-Actor-Critic(SAC) [45] as the reinforcement learning algorithm backbone. Each object is regarded as an agent and can communicate with all the other agents. At each time step $t$, the reward for the i-th agent is $r_t = r_t^{tar} + \lambda * r_t^i$, where $r_t^{tar} = \sum_{1 \leq k \leq t} \langle \mathbf{\Phi}_{tar}(\mathbf{s}_t), \mathbf{s}_k - \mathbf{s}_{k-1} \rangle$ is a centralised target likelihood reward, $\lambda > 0$ is a hyperparameter and $r_t^i = \sum_{j \neq i} c_{i,j}$ is a decentralised collision penalty where $c_{i,j}$ is a collision penalty. Specifically, when a collision is detected between object $i$ and $j$, $c_{i,j} = -1$, otherwise $c_{i,j} = 0$.

## 5 Experiment Setups

### 5.1 Environment

We design two object rearrangement tasks without explicit goal specification for evaluation: *Ball Rearrangement* and *Room Rearrangement*, where the former uses controlled environments for better numerical analysis and the latter is built on a real dataset with implicit target distribution.

**Ball Rearrangement** includes three environments with increasing difficulty: *Circling*, *Clustering*, and the hybrid of the first two, *Circling + Clustering*, as shown in Fig. 3. There are 21 balls of the same geometry in each task. The balls are divided equally into three colours in *Clustering* and *Circling+Clustering*. The goals of the three tasks are as follows: *Circling* requires all balls to form a circle. The circle's centre can be anywhere in the environment; *Clustering* requires all balls to form into three clusters by colour. The joint centre of each cluster has two choices (*i.e.*, red-green-blue or red-blue-green clockwise); *Circling + Clustering* requires all balls to form into a circle where balls of the same type are adjacent. To slightly increase the complexity of the physical dynamics, we enlarge the lateral friction coefficient of each ball and make each ball with different physical properties, *e.g.*, different friction coefficients and masses. The target examples of each task are sampled from an explicit process. Thus, we can define a 'pseudo likelihood' on a given state according to the sampling process to measure the similarity between the state and the target distribution.

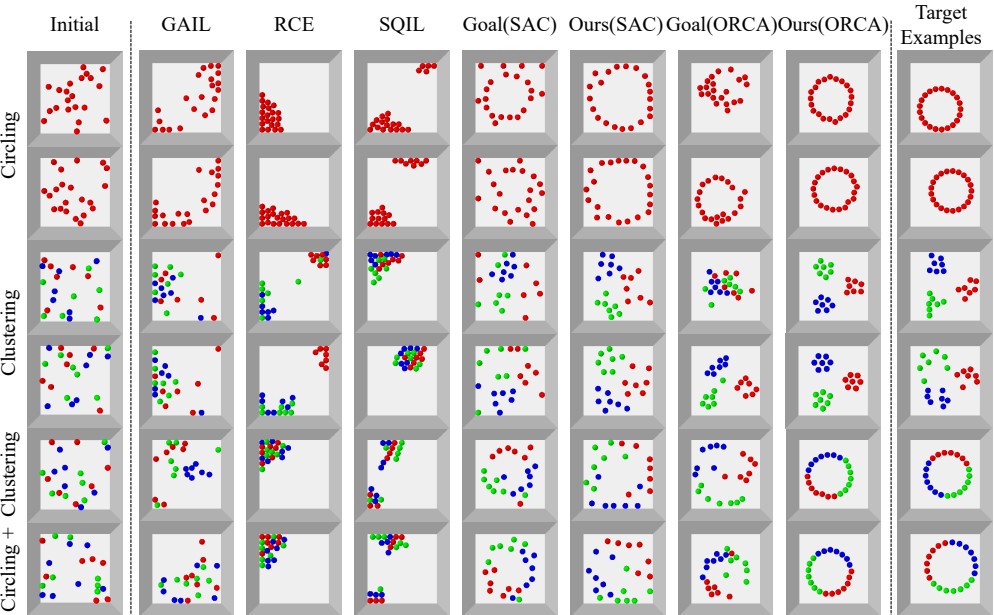

Figure 3: Starting from the same initial states on the far left, we demonstrate the rearrangement results of different methods. On the far right, we demonstrate some target examples of each task.

| GAIL | RCE | SQIL | Goal(SAC) | Ours(SAC) | Target Examples |
|------|-----|------|-----------|-----------|-----------------|

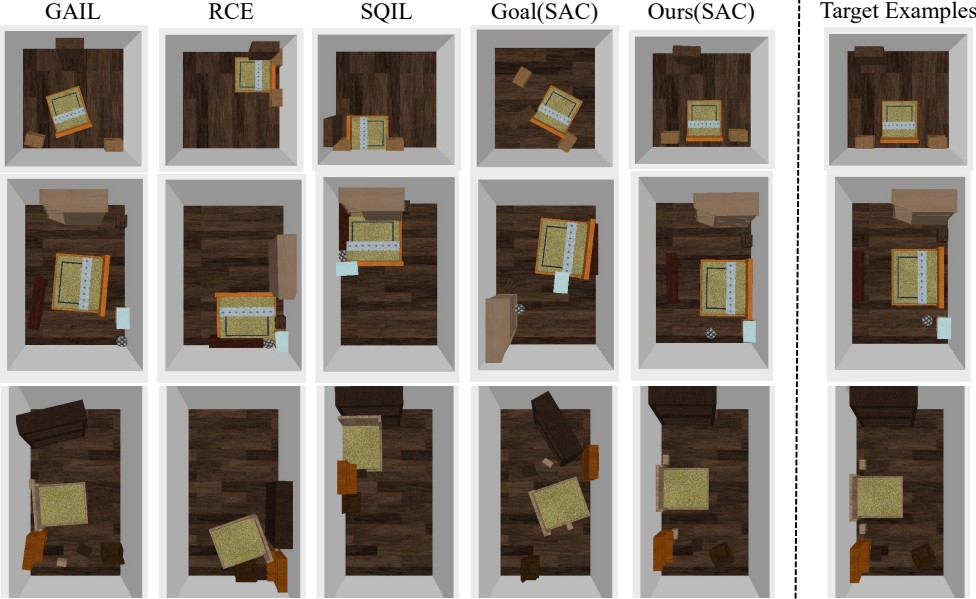

Figure 4: Qualitative results on room rearrangement. We obtain eight rearrangement results of each method starting from the same set of eight initial states. Then we demonstrate the rearrangement result closest to each method's target example.

**Room Rearrangement** is built on a more realistic dataset, 3D-FRONT [46]. After cleaning the data, we get a dataset with 839 rooms, 756 for training, *e.g.*, target examples, and 83 for testing. In each episode, we sample a room from the test split and shuffle the room via a Brownian Process at the beginning to get an initial state. The agent can assign angular and linear velocities for each object in the room at each time step. To tidy up the room, the agent must learn implicit knowledge from the target examples, as shown in Fig. 4. More details are in the Appendix A.

## 5.2 Evaluation

For baseline comparison and ablation studies, we collect a fixed number of trajectories $T = \{\tau_i\}$ starting from the same initial states for each of the 5 random seeds. For each random seed, we collect 100 and $83 \times 8$ (8 initial states for each room) trajectories for ball and room rearrangement, respectively. Then we calculate the following metrics on the trajectories, where the PL is only reported in ball rearrangement. More details of evaluation are in the Appendix D.

**Pseudo-Likelihood (PL)** measures the efficiency of the rearrangement process and the sample quality of the results. We specify a *pseudo-likelihood function* $\mathbf{F}_{proxy} : \mathcal{S} \to \mathbb{R}^+$ for each environment that measures the similarity between a given state and the target examples. For each time step $t$, we report the mean PL over trajectories $\mathbb{E}_{\tau \sim T}[\mathbf{F}_{proxy}(\mathbf{s}_t)]$ and the confidence interval over 5 random seeds. We do not report PL on room rearrangement as it is hard to program human preferences.

**Coverage Score (CS)** measures the diversity and fidelity of the rearrangement results (*i.e.* the terminal states $S_T = \{\mathbf{s}_T\}$ of the trajectories) by calculating the Minimal-Matching-Distance (MMD) [47] between $S_T$ and a fixed set of examples $S_{gt}$ from $p_{tar}$: $\sum_{\mathbf{s}_{gt} \in S_{gt}} \min_{\mathbf{s}_T \in S_T} ||\mathbf{s}_{gt} - \mathbf{s}_T||$. If the rearrangement results of a method miss some modes in the ground truth example set, the coverage score will increase to hurt the performance.

**Averaged Collision Number (ACN)** reflects the safety and efficiency of the rearrangement process since object collision will lead to object blocking and deceleration. ACN is calculated as $\sum_{\tau \in T} \sum_{\mathbf{s}_t \in \tau} \mathbf{c}_t$, where $\mathbf{c}_t$ denotes the total collision number at time step $t$.

## 5.3 Baselines

We compare our framework, *i.e.*, *Ours (ORCA) and Ours (SAC)*, with planning-based baselines and learning-based baselines. Besides, we design heuristic baselines as the upper bounds for ball

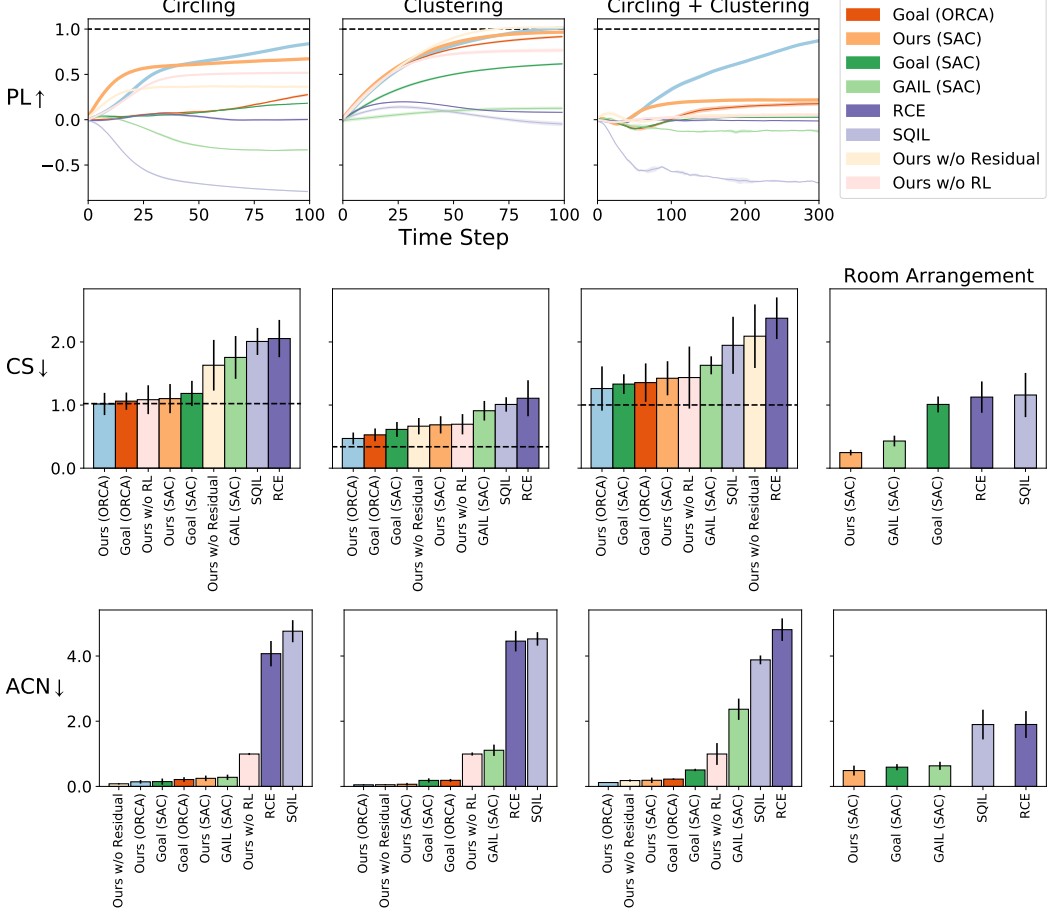

Figure 5: Quantitative results on ball rearrangement. From top to down, we report pseudo-likelihood (PL), coverage score (CS), and averaged collision number (ACN). The PL curves and CS bars of each task are normalised by the oracle's performance of this task. For CS and ACN, we report the mean and confidence interval over five random seeds.

rearrangement tasks. All the reinforcement learning-based methods share the same network design, capacity and hyperparameters. More details of baselines are in Appendix. C.

**Leaning-based Baselines:** *GAIL:* A classical inverse RL method that trains a discriminator as the reward function. *RCE:* A strong example-based reinforcement learning method proposed recently. *SQIL:* A competitive example-based imitation learning method. *Goal-RL:* An 'open-loop' method that first generates a goal at the beginning of the episode via a VAE trained from the target examples and then trains a goal-conditioned policy to reach the proposed goal via RL.

**Planning-based Baselines:** *Goal (ORCA):* This method first generates a goal via the same VAE as in *Goal (SAC)*. Then the agent assigns an action pointing to the goal as reference control and adjusts the preference velocity using ORCA, similar to Sec. 4.3.

**Heuristic-based Baseline:** *Oracle:* We slightly perturb the samples from the target distribution and take these samples as the rearrangement results. This method's rearrangement results will be close to the target distribution yet different from the target examples. Hence, we use this method to normalise the PL curve and CS bars.

# 6 Result Analysis

## 6.1 Baseline Comparison

**Ball Rearrangement:** We present the quantitative results on ball rearrangement tasks in the first three columns of Fig. 5. As shown on the top row of Fig. 5, the PL curves of *Ours (SAC)* and

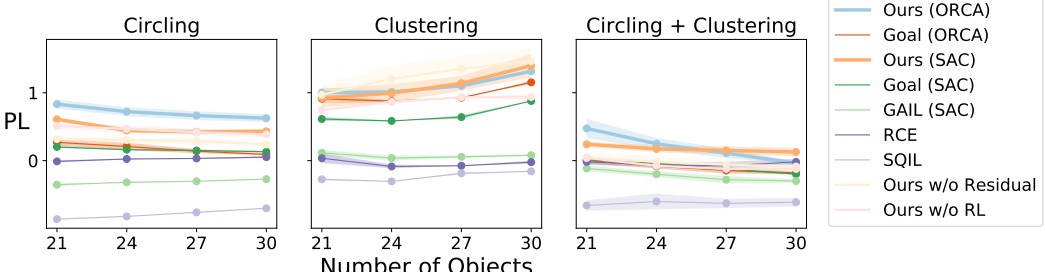

Figure 6: **Scalability Anaslysis**: Each policy is trained on environments with $3 \times 7$ balls and is zero-shot transferred to environments with different ball numbers.

*Ours (ORCA)* are significantly higher and steeper than all the baselines in all three tasks, which shows the effectiveness of our method in arranging objects towards the target distribution efficiently. In the middle row of Fig. 5, *Ours (ORCA)* achieves the best performance in CS across all three tasks. *Goal (SAC)* outperforms *Ours (SAC)* on *Clustering* and *Circling+Clustering* since the goal is proposed by a generative model that can naturally generate diverse and realistic goals. From the qualitative results shown in Fig. 3, our methods, especially *Ours (ORCA)*, also outperform the baselines in realism. Our methods also achieve the lowest ACN in most cases, except that *Goal (SAC)* is slightly better than *Ours (SAC)* in *Circling*.

The *Goal (SAC)* is the strongest baseline yet loses to our framework in efficiency since the generated goals are far from the initial state. We validate this problem by comparing the averaged path length of the *Goal (SAC)* with ours in Appendix. E.3. Results show that our methods converge in a shorter path than goal-based baselines across all three tasks. Besides, the *Goal (SAC)* suffers from invalid goal proposals. As shown in Fig. 3, the *Goal (SAC)* and *Goal (ORCA)* can rearrange objects in the right trend, but the objects may block each other's way. This is because the goals generated by the VAE are not guaranteed to be valid (*e.g.* the balls may overlap with each other), so the agent cannot reach the invalid goal. We demonstrate this problem in Appendix. E.4.

The classifier-based reward learning method, *GAIL*, *RCE*, and *SQIL*, basically fails in most tasks, as shown in Fig. 3 and Fig. 5. This is due to the over-exploitation problem that commonly exists in classifier-based reward learning methods. We validate this in Appendix E.2.

**Room Rearrangement:** We only compare *Ours (SAC)* with learning-based baselines because it is infeasible to employ the ORCA planner in room rearrangement since all the furniture is non-circular. As shown in Fig. 4, our method can obtain more realistic rearrangement results than baselines. As shown in the last column of Fig. 5, the coverage score of *Ours (SAC)* is the lowest and almost half of that of *Goal (SAC)*. Meanwhile, *Ours (SAC)* achieves the lowest ACN compared with baselines. These results prove our method can work on a more complex task and generalise well to unseen conditions.

## 6.2   Ablation Studies and Analysis

We conduct ablation studies on ball rearrangement tasks to investigate: 1) The effectiveness of the *exploration guidance* provided from the target gradient field and 2) The necessity of combining the target gradient field with *control algorithms* to adapt to the environment dynamics. To this end, we construct an ablated version named *Ours w/o Residual* that drops the residual policy learning and another named *Ours w/o RL* that only takes gradient-based action $\mathbf{a}^g$ as policy.

**Ours w/o Residual:** The CS bars in Fig. 5 shows that *Ours (SAC)* significantly outperforms the *Ours w/o Residual* in CS, which means the agent is easy to over-exploit few modes with high density. We also show qualitative results in Appendix E.1 that the objects are usually arranged into a single pattern without the residual policy learning.

**Ours w/o RL:** The PL curves in Fig. 5 show that without the control algorithm serving as the backbone, the rearrangement efficiency of *Ours w/o RL* is significantly lower than *Ours (SAC)* and *Ours (ORCA)*. Meanwhile, the ACN of *Ours w/o RL* is significantly larger than *Ours (SAC)* and *Ours (ORCA)*, which indicates the gradient-based action may cause severe collisions without adapting to the environment dynamics.

**Scalability:** We further evaluate the scalability of our methods by zero-shot transferring the learned model to rearrange different numbers of objects. Specifically, we train all the policies in the ball environment with $3 \times 7 = 21$ balls. During the testing phase, we directly transfer each policy to environments with increased ball numbers, i.e., $3 \times 8$, $3 \times 9$, and $3 \times 10$, without any fine-tuning. In Fig. 6.2, we report the averaged PL increment from each method's initial to target states. Results show that our method still outperforms baselines under different numbers of balls.

## 7 Conclusion

In this study, we first analyse the challenges of object rearrangement without explicit goal specification and then incorporate a target gradient field with either model-based planning or model-free learning approaches to achieve this task. Experiments demonstrate our effectiveness by comparisons with learning-based and planning-based baselines.

**Limitation and Future Works.** This work only considers the planar state-based rearrangement with simplified environments where the agent can directly control the velocities of objects. In future works, it is necessary to extend our framework to more realistic scenarios, *e.g.*, manipulating objects by robots in a realistic 3D environment [11, 48]. We summarise the future directions in the following:

- *Visual Observation:* We may explore how to conduct efficient and accurate score-based reward estimation from image-based examples and leverage the recent progress on visual state representation learning [49, 50, 51, 52] to develop the visual gradient-based action.
- *Complex Dynamics:* When the agent can only move one object at a time, we may explore hierarchical methods where high-level policy chooses which object to move and low-level policy leverages the gradient-based action for motion planning. When we can only impose forces on objects instead of velocities, we may explore a hierarchical policy where a high-level policy outputs target velocities and a low-level employ a velocity controller, such as a PID controller, to mitigate the velocity errors.
- *Multi-Agent Scenarios:* The framework can also be extended to multi-agent scenarios [19, 53, 54, 55], where the decentralised agents need to estimate its gradient field and take actions according to the local observation for multi-agent cooperation or competition.

**Ethics Statement.** Our method has the potential to build home-assistant robots. We evaluate our method in simulated environments, which may introduce data bias. However, similar studies also have such general concerns. We do not see any possible major harm in our study.

## Acknowledgement

This work was supported by the National Natural Science Foundation of China -Youth Science Fund (No.62006006): Learning Visual Prediction of Interactive Physical Scenes using Unlabelled Videos and China National Post- doctoral Program for Innovative Talents (Grant No. BX2021008). We also thank Yizhou Wang, Tianhao Wu and Yali Du for their insightful discussions.

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
