# OpenReview forum: "TarGF: Learning Target Gradient Field to Rearrange Objects without Explicit Goal Specification"
_NeurIPS.cc/2022/Conference — NeurIPS 2022 Accept_

### Official Review · Reviewer_NBX4 · 2022-07-09

**Rating:** 6
**Confidence:** 4
**Soundness:** 3 good
**Presentation:** 2 fair
**Contribution:** 2 fair

**Summary:**

This paper learns a reward function / score function for objective rearrangement tasks, making use of recent advancements in score-based generative modeling. The authors term this score function a “target gradient field”, and show that it can be used with either a path planner or a reinforcement learning algorithm to solve object rearrangement tasks in a toy setting (i.e. where you can directly control object velocities). The authors compare against a number of baselines and competing methods in these simulated object rearrangement domains, and show that their method performs well across several different metrics.


**Questions:**

Some questions

- “learning a score function to qualify a state” → what does “qualify” mean here? Can this sentence be re-written to make it less ambiguous?
- How do you ensure collision avoidance when using model-free RL? Or is collision avoidance not pursued in this setting?


**Limitations:**

I don't think the authors have done a good job of addressing the limitation of the work. In the limitations section, the only limitation that the authors mentioned is the fact that their work only considered planar settings, and not 3D settings. There is only a very brief mention of the fact that future work could use real robots. I think the authors should provide a more detailed limitations section (by reducing space used elsewhere). Some ideas for limitations can be seen in the weaknesses section above. A somewhat more comprehensive plan on how the method could be applied to more realistic settings would also be useful.

**Strengths And Weaknesses:**

Strengths
- Learning reward functions is an interesting application of score-based generative modeling, and has not been explored before, as far as I know.
- Proposed approach performs well against other reward learning approaches on the object-rearrangement task.

Weaknesses
- The paper is somewhat limited in scope -- it is only applied to a very specific robotics problem (that of object rearrangement), and even here some major simplifying assumptions had to be made (such as the fact you can directly control the velocity of any object.
- The paper only shows results in low-dimensional domains (small graphs). Since score-based generative modeling also works in high-dimensional domains (such as images), it would be interesting to see if the method can be used for reward learning from scene images, for example.
- While the paper is nicely structured, it contains a large number of typos and grammatical errors. These errors must be fixed before the paper is ready for publication. I will provide a non-exhaustive list here, but I would also suggest making useful of professional proof-reading services if possible.

Typos and grammatical errors:
- “We do also tries to find a example-based control method”
- “Differently, our proposed approach focus on learning the target gradient fields from the examples.”
- “Besides, both RL method and traditional planner can be supported by our target gradient fields in the object arrangement task.”
- “even if we are accessible to the target distribution”
- “In address these problems”
- "reducing the risk of objects being collided”

---

> ### Author Response · Authors · 2022-08-02
> **Thank you! We have clarified some misunderstandings, address your concerns, and hope to hear back from you if you have further questions! [part 2/2]**
>
> >**Q6**: How do you ensure collision avoidance when using model-free RL?
>
>
> **A6**: As mentioned in L202 in the main paper, the RL agent will receive a centralised reward (likelihood) from the score network and a decentralised reward (collision penalty $c_t^i $) from the environment.
>
> Hence, the reward for the i-th agent at timestep t can be written as $r_t^i = r_{likelihood} - \lambda*c_t^i$. As described in supplementary Sec. 2.4(and 3.1), $\lambda > 0$ is a hyper-parameter to balance the immediate reward and the collision penalty, the collision penalty counts the total number of collisions to agent i $c_t^i = \sum_{j\neq i} col_{i, j}$, where $col_{i, j}$ equals to 1 when i-th and j-th agent collide with each other and 0 elsewhere.
>
> Thank you for pointing this out. We revised the Sec. 4.4 of our main paper accordingly to make this clear.
>
> >**Q7**: I don't think the authors have done a good job of addressing the limitation of the work...
>
> **A7**: Thanks for the suggestions. We have updated the limitation section of our paper.
>
> We sincerely hope that our response above makes things clearer to you and addresses your concerns well. Otherwise, please do not hesitate to ask us more, and we are very happy to discuss further. Thank you again for the useful comments and questions!
>
> [1]Discovering Generalizable Spatial Goal Representations via Graph-based Active Reward Learning. Aviv Netanyahu*, Tianmin Shu*, Joshua B. Tenenbaum, and Pulkit Agrawal. ICML 2022

---

> ### Author Response · Authors · 2022-08-02
> **Thank you! We have clarified some misunderstandings, address your concerns, and hope to hear back from you if you have further questions! [part 1/2]**
>
> > **Q1**: The paper is somewhat limited in scope. It is only applied to a very specific robotics problem (that of object rearrangement)...
>
> **A1**: Object arrangement is not limited to the room scenario. For instance, in multi-agent formation control, the UAV/UGV are required to move together to form a pattern in the shortest path. Our ball arrangement tasks are exactly inline with this real-world scenario.
>
> The reason why we further evaluate our method in a room scenario is that we would like to show our method can handle more observational variables, e.g., orientation, object size, and category. Inspired by the multi-chair arrangement example in our demo video, we enable all objects to be moveable to simplify the dynamics so as to emphasize the key difficulty of the arrangement task.
>
> **Object arrangement task is an underexplored problem.** To emphasize the key difficulty of this problem, we evaluate our method in controlled environments with fewer variants. This research paradigm is common in machine learning communities. Also, a concurrent work for arrangement study in ICML [1] (released after our submission) adopts a similar research paradigm. Compared with this work, our experiment setting considers significantly more objects with diverse attributes (e.g. categories, bounding box).
>
>
> >**Q2**: Here, some major simplifying assumptions had to be made (such as the fact you can directly control the velocity of any object ...
>
>
> **A2**: All comparisons with the baseline use the same action space and information, so this point would not affect our conclusion in terms of system design. For the force-based action, we can easily extend the velocity-oriented policy with a two-level hierarchy controller, i.e., the high-level controller outputs the expected velocity, and the low-level controller (a PID-like controller) outputs the force to adjust the velocity accordingly.
>
> To evaluate the effectiveness of this hierarchy controller, we further conduct experiments on the clustering + circle environment. We compare this extended bi-level approach with Ours(SAC) in our paper. The results in Sec. 4 in [**our site**](https://sites.google.com/view/neurips2022-paper2108-rebuttal/)  show that *velocity-oriented gradients remain meaningful to force-based action space* but will face some cost in time steps due to the control error.
>
>
> >**Q3**: The paper only shows results in low-dimensional domains (small graphs). Since score-based generative modelling also works in high-dimensional domains (such as images), it would be interesting to see if the method can be used for reward learning from scene images, for example.
>
>
> **A3**: Thank you for this question. To show our method can be used for reward learning from raw-pixel images, we further analyse our framework by training a target score network that takes the image as input and then use the trained target score network to train our policies.
>
> We compare this image-based gradient field (denoted as Ours(Image)) with 1) the state-based target score network, denoted as *Ours(State)*; and 2) a goal-conditioned baseline, denoted as *Goal(State)*.
> Note that *Ours(State)* and Goals(State) represent *Ours(SAC)* and  *Goal(SAC)* in the main paper, respectively.
>
> Results in [**our site**](https://sites.google.com/view/neurips2022-paper2108-rebuttal/) Sec. 5 show that *Ours(Image)* achieves comparable results with Goal(State), demonstrating our framework still has good performance even though the target score network uses the raw-pixel images as input.
> These results also indicate that raw-pixel observation is a distractor compared with our key focus. Hence, we choose to conduct our experiments in a state-based setting in our main paper.
>
>
> >**Q4**: While the paper is nicely structured, it contains a large number of typos and grammatical errors.
>
>
> **A4**: Thanks for pointing this out. We have fixed most typos and grammatical errors in the revision.
>
>
> >**Q5**: “learning a score function to qualify a state” → what does “qualify” mean here?
>
> **A5**: Thanks for pointing this out. Here we mean learning a score function to quantify the arrangement likelihood of a state. We have revised the paper accordingly.

---

> ### Comment · Reviewer_NBX4 · 2022-08-04
> **Response to author response.**
>
> Thank you for the response! I don't have any further questions at the moment. Given the response, I have updated my rating to 6.

---

> > ### Author Response · Authors · 2022-08-04
> > **Thank you**
> >
> > Thanks for raising your rating to 6. We are glad that our responses help alleviate your concerns. Thanks again for all your valuable feedback!

---

### Official Review · Reviewer_ZnF9 · 2022-07-11

**Rating:** 6
**Confidence:** 4
**Soundness:** 3 good
**Presentation:** 2 fair
**Contribution:** 3 good

**Summary:**

## Post Rebuttal

I thank the authors for their response. I have increased my score.

## Pre Rebuttal

This paper attempts to tackle the problem of learning to rearrange a set of objects to a "sensible" state without a pre-defined reward function. Towards this end, they propose Target Gradient Fields (TarGF). TarGF estimates the gradient of the likelihood that the current environment state is a goal (or "sensible") state for the objects.

TarGF is learned by randomly permuting correct arrangements.

The output of TarGF is used to estimate the best action to take at every timestep and/or used to estimate the reward.

The method evaluated in two scenarios. In the first a set of balls must be rearranged into either a circle, clusters by color, or a combination. In the second the furniture in rooms must be rearranged back to its initial states.

**Questions:**

Is (1) a well-defined optimization problem? For any non-trivial starting state, won't p_tar(s_0) always be negative infinity, so the discounted sum will always be negative infinity

Comments:
Rearrangement should be used instead of arrangement to be inline with what the community is calling this task. See Batra et. al. 2020.

**Limitations:**

Limitations are addressed.

**Strengths And Weaknesses:**

# Strengths
The proposed method is able to learn solely from examples and does not require any additional data other than correct examples (negatives are generated by adding noise to the correct examples).

In the circle scenario, the proposed method outperforms various baselines across a series of metrics -- namely, collision count and success.

# Weaknesses

The evaluation is very simple. There isn't an agent that rearranges these objects, instead the objects all rearrange themselves. This is a good initial evaluation that shows the method provides a reasonable reward signal, but it leaves a lot of questions since this reduces the time-horizon, isn't representative of how the objects would be moved by a single agent (one object moves at a time), and removes the initial exploration to find the objects. An evaluation that is closer to a real scenario would be helpful.

Ideally, the authors would evaluate on something like HAB (Szot et al 2021), or AI2Thor rearrangement.

The residual policy derived from the gradient-based action seems quite important to overall performance of the model free approach. However, this residual policy requires an oracle entity-based representation of the state -- requires all object positions, object categories, and bounding boxes. While it is fine to assume this at training time, this is also required at evaluation time. I am concerned by the assumption that such a representation is available as the task complexity increases.

On this topic, it is unclear if the baselines receive the same information. The supplement says they operate on state, but state is defined just as object position, not position + category + bounding box, in the paper.

Finally, in the most realistic scenario, Room Arrangement, the proposed method likely doesn't outperform GAIL + SAC by a statistically significant margin. Can the authors report confidence intervals instead of variance?

---

> ### Author Response · Authors · 2022-08-02
> **REPLY: Thank you! We have clarified some misunderstandings, address your concerns, and hope to hear back from you if you have further questions![part 2/2]**
>
> > **Q4**: Is (1) a well-defined optimization problem? For any non-trivial starting state, won't p_tar(s_0) always be negative infinity, so the discounted sum will always be negative infinity.
>
> **A4**: Thank you for this valuable question! We can additionally assume that the $p_{tar}(s_0)$ always be positive.
> If there are some states that  $p_{tar}(s_0) =0$, we can slightly perturb all the original target examples using a small Gaussian noise (e.g., N(0, 0.0001)). Then we can replace the original target distribution with this perturbed one at almost no cost. The above trick used in our implementation was previously used in [5] to tackle the manifold hypothesis issue. We will add the details in the revision.
>
> >**Q5**: This reduces the time horizon, isn't representative of how the objects would be moved by a single agent (one object moves at a time).
>
> **A5**: As described in the previous question, we argue that multiple objects moving together is also a practical setting. We agree there are cases where objects should be moved one by one, but our framework still has the potential to meet this setting. To this end, we design a bi-level approach (denoted as *Ours + Planner*) for object arrangement: The high-level policy determines which object to move according to the trained target score network (e.g. choosing the object with the largest gradient component). The low-level policy leverages the target score network and ORCA planner to output the action.
>
> We compare this approach with another heuristic-based bi-level planner (denoted as *Goal + Planner*): The high-level planner first generates goals for each object and chooses the object with the farthest distance to the goal to move. The low-level planner is the same as *Ours + Planner*.
>
> As shown in Figure. 3 in  [**our site**](https://sites.google.com/view/neurips2022-paper2108-rebuttal/), *Ours + Planner* is better than *Goal + Planner* in efficiency. This shows the effectiveness of our methods in handling the scenario where the agent can move one object at a time.
>
> > **Q6**: However, this residual policy requires an oracle entity-based representation of the state ...I am concerned by the assumption that such a representation is available as the task complexity increases.
>
> **A6**: Thank you for pointing this out. Currently, we focus on the setting where the state is fully observable to the agent. Hence, the oracle states of the objects are available to the agent. Considering a more complex setting, e.g., vision-based control, there are three ways to realize a vision-based object arrangement:
>
> - 1)  employ the computer vision models to extract the explicit state from the image, e.g., using rotated box detection[1] to provide the bounding box and category of objects.
> - 2) Learn an object-centric world model as an implicit representation of the oracle state [2].
> - 3) Learning the vision-based policy by cloning the state-based behaviour, such as [3].
>
> > **Q7**: It is unclear if the baselines receive the same information ... not position + category + bounding box, in the paper.
>
> **A7**: Thank you for pointing this out. All the baselines and our methods receive the same state representation in the same task, yet the representations differ in different tasks.
>
> - For ball arrangement, the detailed state representations are mentioned in Supp 1.1.
> - For room arrangement, the state is defined as the concatenation of 2-D position, 1-D orientation, 2-D bounding box, and a category label. We added the details in the supplementary Sec. 1.2.
>
>
> > **Q8**: Rearrangement should be used instead of arrangement to be inline with what the community is calling this task. See Batra et. al. 2020.
>
> **A8**: The problem setting we studied is different from *Batra et. al. 2020* in motivation, formulation, and challenges. As described above, the room rearrangement focus on the planning and exploration problems in embodied AI, where the goal is given and deterministic. Instead, the core challenge of object arrangement is **learning to control with examples and without reward**. So we do not agree that rearrangement should be used in this work.
>
> We sincerely hope that our response above makes things clearer to you and addresses your concerns well. Otherwise, please do not hesitate to ask us more, and we are very happy to discuss further. Thank you again for the useful comments and questions!
>
> **Reference:**
> [1] https://github.com/open-mmlab/mmrotate
>
> [2] Kipf, Thomas, Elise van der Pol, and Max Welling. "Contrastive learning of structured world models." ICLR 2020.
>
> [3] Zhong, Fangwei, et al. "Towards distraction-robust active visual tracking." International Conference on Machine Learning. ICML 2021.
>
> [4] Netanyahu, Aviv, et al. "Discovering Generalizable Spatial Goal Representations via Graph-based Active Reward Learning." ICML 2022.
>
> [5] Yang Song and Stefano Ermon. Generative modeling by estimating gradients of the data distribution. NeurIPS 2019 (Oral).

---

> > ### Comment · Reviewer_ZnF9 · 2022-08-08
> > **Reply**
> >
> > I thank the authors for their response. The planner results in particular are really quite compelling as it both shows the efficacy of the learned reward in a longer horizon setting and shows that the reward is stable when only a single object is being moved at a time. The updated limitation section is also appreciated.
> >
> > The only point I will push back on is room rearrangement vs object arrangement. While Batra et al didn't describe the author's object arrangement exactly as one of their examples and the present challenge of object arrangement and room rearrangement are different (IRL and RL, respectively), they are very similar tasks as the end of the day; the goal specification is just different. [1], [2], [3] all study similar tasks and refer to their task as rearrangement (note that these are concurrent works). My encouragement of using rearrangement isn't to detract from the novelty of this work, but to make sure this work can be properly appreciated by community working on a very similar task and so that it's easier for readers to understand how this work relates to others.
> >
> >
> > [1] Netanyahu, Aviv, et al. "Discovering Generalizable Spatial Goal Representations via Graph-based Active Reward Learning." ICML 2022.
> > [2] Yash Kant et al, "Housekeep: Tidying Virtual Households using Commonsense Reasoning", ECCV 2022
> > [3] Gabriel Sarch et al, "TIDEE: Tidying Up Novel Rooms using Visuo-Semantic Commonsense Priors", ECCV 2022.

---

> > > ### Author Response · Authors · 2022-08-08
> > > **REPLY: Many thanks again for your response! We hope to hear back if you have further questions!**
> > >
> > > > I thank the authors for their response. The planner results in particular are really quite compelling as it both shows the efficacy of the learned reward in a longer horizon setting and shows that the reward is stable when only a single object is being moved at a time. The updated limitation section is also appreciated.
> > >
> > > Many thanks for your response. We are glad that our responses help alleviate your concerns.
> > >
> > > > The only point I will push back on is room rearrangement vs object arrangement. …..
> > >
> > > For this only point, we appreciate you for reminding us of these concurrent works. As described in Sec 2.1 of our main paper, there are also a number of previous works naming similar tasks as "Object arrangement", particularly in computer graphics and robotics. Nevertheless, we acknowledge that the two papers in ECCV 2022 (The acceptance is announced on Jul. 3) also name the setting of "manipulating objects without explicit specifying goal" as "rearrangement". **So, we have renamed our "arrangement" task to "rearrangement", as requested.**
> > >
> > > These concurrent works also demonstrate that the problem we study is important and challenging to the community. We are also glad to see these newly released datasets, which can help us further extend our method and conduct experiments on the embodied AI scenarios. **We would like to extend our framework to these concurrent benchmarks in the future.**
> > >
> > > We would like to emphasise that we are "rearranging objects without explicit goal specification" rather than achieving a specific goal. The concurrent works you mentioned [2, 3] exploit the commonse knowledge from Large Language Model (LLM) or memex graph to infer rearrangements goals. In this paper, we focus on a more fundamental problem in rearrangement: *how to estimate the similarity (i.e., the target likelihood) between the current state and the example sets and manipulate objects to maximise it*. We only need a set of positive state examples to learn the gradient field, rather than the prior knowledge about specific scenarios. **We revised our main paper accordingly and described the differences between our work and the others** to ensure our work can be properly appreciated by the community working on a similar task.
> > >
> > > Many thanks again for your response! We hope to hear back if you have further questions!

---

> > > ### Author Response · Authors · 2022-08-09
> > > **Would you update your rating?**
> > >
> > > We would like to ask whether we have addressed your concerns for **updating your rating**? Please let us know whether you have further questions. We are sincerely waiting for discussion with you

---

> ### Author Response · Authors · 2022-08-02
> **REPLY: Thank you! We have clarified some misunderstandings, address your concerns, and hope to hear back from you if you have further questions![part 1/2]**
>
> > **Q1**: The evaluation is very simple ... (one object moves at a time).
>
> **A1**: Sorry for making you confuse our object arrangement with the existing room rearrangement.  We argue that the rearrangement and arrangement are two quite different tasks, even though their names are similar.
>
> For a clear comparison, we compare the two tasks in the following table in different aspects:
>
> |                    | Room Rearrangement | Object Arrangement |
> |--------------------|--------------------|-------------|
> | Deterministic Goal 	| Yes                | No          |
> |Pre-defined Reward  	| Yes                 |    No           |
> | Observation       		 | Partial		| Global      |
> | Learning Paradigm    |    Reinforcement Learning  |  Inverse Reinforcement Learning |
>
> **We highly encourage you to read the [*common responses*](https://openreview.net/forum?id=Euv1nXN98P3&noteId=jlgfVD7Hhq) for a better understanding of the difference between the two tasks.**
>
> We agree that evaluating the methods in a realistic 3D environment can further improve our paper. But such 3D environments will bring additional distractors (partial observation, dynamic noises) that make the results too complex to analyze, particularly at the current stage.  Note that a concurrent work in ICML 2022 [4] in object arrangement, published after our submission, also uses 2D environments for evaluation.
>
> **Object arrangement is not limited to the room scenario.** For instance, in multi-agent formation control, the UAV/UGV are required to move together to form a pattern in the shortest path. Our ball arrangement tasks are exactly inline with this real-world scenario.
>
> The reason why we further evaluate our method in a room scenario is that we would like to show our method can handle more observational variables, e.g., orientation, object size, and category. Inspired by the multi-chair arrangement example in our demo video, we enable all objects to be moveable to simplify the dynamic, so as to emphasize the key difficulty of the arrangement task.
>
> **Object arrangement task is an underexplored problem.** To emphasize the key difficulty of this problem, we evaluate our method in controlled environments with fewer variants. Note that a concurrent work for reward learning in ICML [4] (released after our submission), adopts a similar research paradigm. Compared with this work, our experiment setting considers significantly more objects with diverse attributes (e.g. categories, bounding box).
>
> We have refined our paper to make it more clear.
>
>
> >**Q2**: In the second the furniture in rooms must be rearranged back to its initial state.
>
> **A2**: Actually, we do not require rearranging the furniture back to its initial state.
> In Figure 4, Ours(SAC) arrangement results look similar to the 'ground truth' rooms because we demonstrate **the nearest configuration** w.r.t. its 'ground truth'.
> Thank you for pointing this out. We revised the caption of Figure 4 of our main paper accordingly to make this clear.
>
> > **Q3**: ... likely doesn't outperform GAIL + SAC ... Can the authors report confidence intervals instead of variance?
>
> **A3**: Thank you for pointing this out, we have replaced the variance intervals on paper with confidence intervals. It is clear that our method outperform GAIL+SAC with a large margin in room arrangement (Ours: 0.040 +- 0.002 vs. GAIL: 0.173 +- 0.007).

---

> ### Author Response · Authors · 2022-08-09
> **Thank you!**
>
> Thanks for raising your rating to 6. We are glad that our responses help alleviate your concerns. Thanks again for all your valuable feedback!

---

### Official Review · Reviewer_eUMp · 2022-07-11

**Rating:** 7
**Confidence:** 4
**Soundness:** 3 good
**Presentation:** 4 excellent
**Contribution:** 3 good

**Summary:**

This work introduces a novel approach to tackling object rearrangement. The main idea is to learn a scoring function in the form of a gradient field. The learned gradient field could be used by a model-based planning algorithm to find collision-free arrangement plans or as a reward function to train RL agents. Experiment results show promising results of the proposed method for rearranging a room and sample efficient policy learning.

**Questions:**

1. When talking about collision-free room arrangement, do we consider the 2d rotation of the furniture during the movements? Basically, what’s the action space, and why it is hard compared to the ball arrangement?
2. Did you test the generalization of the proposed method? e.g. Testing in an unseen room with novel furniture.

**Limitations:**

1. Might fail to handle multi-model goal distributions.
2. Missing some details of the experimental setups in the main paper.

**Strengths And Weaknesses:**

Strength:
1. The paper is well-written with a clear introduction of the implementation details.
2. I really enjoy reading the paper following the motivation of how the learned gradient field could be used to tackle three major challenges of object arrangement, which is easy to follow and makes a lot of sense.
3. The velocity-based action space of the designed task showcases that the proposed method could be used in complex continuous control scenarios. The discussion of collision avoidance planning with ORCA further strengthens the practicality of the proposed approach in real-world tasks.
4. It is impressive to see such a gradient-based scoring function could be used as a reward function for training the RL agent. Also, the gradient-based action could further alleviate the burden of the action generation network so that only a small residual policy network needs to be trained, which largely improves the sample efficiency of the policy learning process.

Concerns:
1. How does the RL residual policy learn to avoid collision? As mentioned in the model-based planning method, the gradient field does not have environment information so the gradient cannot avoid collision. Since the residual policy is based on gradient-based action, does it need an additional collision penalty reward during training? In the paper, you mentioned the centralized and decentralized reward but more details here would be better.
2. The gradient field might struggle to handle multi-model distributions. What if there are multiple arrangement goals for the same environment. Will the gradient become a mean of different distributions which will mislead the planner or the policy learning? I feel a probabilistic model would make more sense for this situation.
3. The authors mentioned that the room-arrangement task is too complex for planning-based methods. I would suggest trying with bi-level planning which first plans high-level arrangements plans with the learned gradient and then use a collision-avoidance trajectory planner to optimize the path to avoid the collision.

---

> ### Author Response · Authors · 2022-08-02
> **REPLY: Thank you! We have clarified some misunderstandings, address your concerns, and hope to hear back from you if you have further questions![part 2/2]**
>
> > **Q4**: I would suggest trying with bi-level planning which first plans ... to avoid the collision
>
> **A4**: Thanks for your insightful suggestion!
> In fact, the Goal (ORCA) baseline in ball arrangement can be regarded as an implementation of bi-level control. We do not apply it in the room arrangement as the rectangle-shape furniture objects do not satisfy the circle-shape assumption made by the decentralised planner ORCA. We also tried to implement a centralised planning algorithm (e.g. RRT [1]). However, it is costly in time to search for a reasonable path, i.e., taking almost 1 min for 3x3 balls. If we increase the number of objects (i.e., from 3x3 to 3x7 balls), we find that it failed to search a motion path in a limited time (i.e. 10 minutes).
>
> We notice that such a solution is of two main limitations:
> 1. Open-loop planning:  The proposed goal and the initial state may not be reachable or far away from each other.
> 2. Accessibility of the generated goal: The goal proposer ignores the environment dynamics. So the generated goal may be physically inaccessible (e.g. objects overlap with each other), as shown in Fig.4 in the supplementary.
>
> As demonstrated in the ball arrangement experiments, the above two limits lead to weak performance :
> 1. In Fig. 5 of the main paper, the likelihood curves of goal-based methods are significantly below ours. In Table 1 of the supplementary, the average length of trajectories of goal-based methods is significantly below ours (e.g., In Circling + Clustering, the averaged state change of Ours(SAC) 48.93 +- 4.68 achieves less than half of Goal(SAC) 122.72 +- 5.93).
> 2. Besides, in the room arrangement, the rectangle-shape furniture objects do not satisfy the circle-shape assumption made by the decentralised planner ORCA. We also tried to implement a centralised planning algorithm for ball arrangement (i.e. RRT). However, though this algorithm can find a feasible solution in one minute when the object number is less than 3*3, it failed to search a motion plan in ten minutes when the number of objects got higher. This is also the well-known curse of the dimensionality problem of the centralised planner.
>
> These aspects cause unsatisfied performance for the goal-based approach. Thanks for bringing up this baseline. We are willing to add it to the revised paper if requested.
>
> > **Q5**: Did you test the generalisation of the proposed method? e.g. Testing in an unseen room with novel furniture.
>
> **A5**: Yes, we tested the generalisation.
> In the ball arrangement, we test the generalisation of the unseen initial state across various numbers of balls. To be specific, the gradient fields and policy are trained in the environment with 3x7 balls. Then, we test the learned policy in the environment with 3x8, 3x9, and 3x10 balls. The results are shown in Fig.6 of the main paper.
>
> In the room arrangement, we emphasise evaluating the generalisation of the target score network. The target score network is trained under 756 room examples and tested on 83 unseen environments. In this case, we train the RL-based policy in 83 testing environments with the pre-trained target score network, which provides the reward and gradient-based action for policy learning.
>
> To further evaluate the generalisation of the learned policy, we conduct an additional experiment, where we also train the policy in 756 rooms and evaluate the policy in 83 unseen environments. The results of the above setting are reported as below:
>
> | Setting                          | Coverage Score | Collision Num  |
> |----------------------------------|----------------|----------------|
> | Gradient(Unseen), Policy(Seen)   | 0.038 +- 0.001 | 0.152 +- 0.007 |
> | Gradient(Unseen), Policy(Unseen) | 0.041 +- 0.002 | 0.145 +- 0.002 |
>
> We can see that the policy trained in the different settings are of comparable performance, showing the good generalisation of our method in room arrangement.
>
> We sincerely hope that our response above makes things clearer to you and addresses your concerns well. Otherwise, please do not hesitate to ask us more, and we are very happy to discuss further. Thank you again for the useful comments and questions!
>
> [1] Rapidly-exploring random trees: A new tool for path planning, LaValle, Steven M and others, 1998

---

> ### Author Response · Authors · 2022-08-02
> **REPLY: Thank you! We have clarified some misunderstandings, address your concerns, and hope to hear back from you if you have further questions![part 1/2]**
>
> >**Q1**: How does the RL residual policy learn to avoid collision?
>
> **A1**: As mentioned in L202 in the main paper, the RL agent will receive a centralised reward (likelihood) from the score network and a decentralised reward (collision penalty $c_t^i$) from the environment. Hence, the reward for the i-th agent at timestep t can be written as $r_t^i = r_{likelihood} - \lambda*c_t^i$. As described in supplementary Sec. 2.4(and 3.1),  $\lambda > 0$ is a hyper-parameter to balance the immediate reward and the collision penalty. The collision penalty counts the total number of collisions to agent i $c_t^i = \sum_{j\neq i} col_{i, j}$, where $col_{i, j}$ equals to 1 when i-th and j-th agent collide with each other and 0 elsewhere.
> Thank you for pointing this out. We revised the Sec. 4.4 of our main paper accordingly to make this clear.
>
>
> >**Q2**: The gradient field might struggle to handle multi-modal distributions.
>
> **A2**: The learned gradient field can handle the multi-modal goal distributions, leading the state to the closest mode. Empirical results on ball arrangement demonstrated supported this argument: The target examples for clustering were sampled from a bi-modal Gaussian distribution. The two modes were of different colour order, i.e., the centres of different clusters ordered in R-G-B or R-B-G. As shown in Fig. 3 of the main paper, images of two middle rows on the rightest column demonstrate these two modes.
> Moreover, we also conduct additional analysis to further validate the effectiveness of our method in cases of multi-modal goal distribution. Specifically, we extend the target distribution of the clustering task to a multi-modal Gaussian and evaluate our method on this task, where the six modes are:
>
> |       | Mode1 | Mode2 | Mode3 | Mode4 | Mode5 | Mode6 |
> |-------|-------|-------|-------|-------|-------|-------|
> | Area1 | R     | R     | B     | B     | G     | G     |
> | Area2 | G     | B     | G     | R     | B     | R     |
> | Area3 | B     | G     | R     | G     | R     | B     |
>
> We also visualise the typical examples of each mode in [**site**](https://sites.google.com/view/neurips2022-paper2108-rebuttal/). Qualitative results demonstrate the learned gradient can successfully guide both planning-based and learning-based planners to reach one mode (instead of a `mean' mode) according to the initial state.
> Besides, we evaluate the latent distribution (i.e., the probability of a state belonging to one specific mode) of the arrangement results (states). In Table. 5, the average entropies of the latent distributions of our methods are lower than GT (target examples). This shows the arrangement results of our methods are closer to the mode centres. The averaged latent distribution (overall arrangement results) of our methods achieve comparable orders of magnitude in different modes. This shows the arrangement of our methods can cover all the mode centres.
>
> >**Q3**: Do we consider the 2d rotation of the furniture during the movements? Basically, what's the action space, and why is it hard compared to the ball arrangement
>
> **A3**: Yes, we do consider the 2d rotation of the furniture during the movements. (See video 3:25-3:35 )
> In the room arrangement, the action space of each object(agent) is a three-dimensional continous vector $(v_x, v_y, v_{yaw})$, where $v_x $ and $v_y$ are the velocity in x-axis and y-axis respectively, $v_{yaw}$ is the angular veloicty.
> We added the details of the state and action spaces of room arrangement in supplementary Sec. 1.2.

---

> ### Author Response · Authors · 2022-08-09
> **We are looking forward to your response!**
>
> As the author-reviewer discussion is ending, we would like to ask whether our replies have addressed your concerns. Please let us know whether you have further questions. **We are sincerely waiting for your response.**

---

> > ### Comment · Reviewer_eUMp · 2022-08-09
> > **Thank you for clarification**
> >
> > I thank the authors' detailed clarification. Most of my concerns are addressed for example the multi-model distribution and experiment task setups. Based on that, I'll increase my score from 6 to 7.

---

> > > ### Author Response · Authors · 2022-08-10
> > > **Thank you!**
> > >
> > > Thanks for raising your rating to 7. We are so glad that our responses help address your concerns. Thanks again for all your valuable feedback!

---

### Author Response · Authors · 2022-08-02
**Common responses to all reviewers**

We thank all reviewers for appreciating our ideas and experiments. "The paper is well-written with a clear introduction of the implementation details."(eUMp), "the learned gradient field could be used to tackle three major challenges of object arrangement, which is easy to follow and makes a lot of sense." (eUMp), "The velocity-based action space of the designed task showcases that the proposed method could be used in complex continuous control scenarios." (eUMp), "The proposed method is able to learn solely from examples and does not require any additional data." (ZnF9), "Learning reward functions is an interesting application of score-based generative modeling, and has not been explored before." (NBX4).

However, we notice that some reviewers may misunderstand our problem setting, and confuse our object arrangement with the room rearrangement. We summarise the difference between these two tasks in the following table:
|                    | Room Rearrangement | Object Arrangement |
|--------------------|--------------------|-------------|
| Deterministic Goal 	| Yes                | No          |
|Pre-defined Reward  	| Yes                 |    No           |
| Observation       		 | Partial		| Global      |
| Learning Paradigm    |    Reinforcement Learning  |  **Inverse** Reinforcement Learning |

- In the room rearrangement setting, the agent aims to rearrange a room into a specific target configuration. If the global state is given, this task is trivial for goal-conditioned reinforcement learning. Hence, in the room rearrangement setting, only partial (visual) observation instead of a global state is provided. This reveals the key difficulties of the room rearrangement: *How to effectively explore the room to infer the rearrangement goal while recognizing and manipulating the objects from visual observation*.
- In our object arrangement setting, the agent **has not specified a goal**. Instead, the agent is *given a set of target examples to infer the arrangement pattern and then moves the objects to increase the target likelihood as efficiently as possible*.
- Hence, they focus on **different research areas**. The room rearrangement focus on the embodied ai, particularly in **3D scene understanding and exploration**. Differently, the object arrangement focuses on **goal distribution inference**, i.e., finding the most efficient path in the physical world to reach a goal distribution. In other words, in room rearrangement, the task is to "make the room the same as a specific reference room". However, in room arrangement, the task is to make the room of similar features to the examples.
- For example, in the case of "tidy up a room", the rearrangement agent requires **a tidied-up room as a reference**, which should be of the same objects (numbers, category, and shape) as the current state and only some objects are of different poses. Hence, we have to prepare a reference room for the rearrangement agent, and the agent has to explore the difference between the two rooms first.  Instead, the arrangement agent only needs **some examples of the tidied-up room during training**. Note that the examples can be collected from other scenes. Hence, there would be diverse states that are expected in object arrangement, but only one expected state is in room rearrangement.
- Moreover, the learning paradigms for these two tasks are also different. Rearrangement policy can be trained via **reinforcement learning (RL)**. That is because we can easily define a reward function for learning as the oracle goal state is specific and known in room rearrangement. However, RL is infeasible in arrangement, since it is hard to define the reward. Hence, the arrangement agent should be trained via **inverse reinforcement learning (IRL)**, i.e., learning a reward from the examples for policy learning, which remains challenging to the community.

Object arrangement is not limited to the room/ desktop scenario. For instance, in multi-agent formation control, the mobile agents (UAV/ UGV) are required to move together to form a pattern in the shortest path. Our ball arrangement tasks are exactly inline with this real-world scenario.

The reason why we further evaluate our method in a room scenario is that we would like to show our method can handle more observational variables, e.g., orientation, object size, and category. Inspired by the multi-chair arrangement example in our demo video, we enable all objects to be moveable to simplify the dynamic, so as to emphasize the key difficulty of the arrangement task.

---

> ### Author Response · Authors · 2022-08-02
> **Additional Experiments**
>
> We also notice that there are some questions and suggestions for our work. We further conduct additional experiments and analyses to demonstrate:
> - 1) Our method can tackle the multi-modal distribution and would not arrange objects into a 'mean pattern'
> - 2) Our proposed score-based reward learning method can work on image-based observation
> - 3) Our method can handle the cases when only one object can move at a time or the agent can only apply force on objects.
>
> We show the vivid results at a [anonymous site](https://sites.google.com/view/neurips2022-paper2108-rebuttal).
>
> We sincerely hope our work contributes to the ML+Robotic research field. Below we reply to reviewers’ questions point-by-point. Thanks again for your valuable comments and suggestions.

---

### Author Response · Authors · 2022-08-04
**Revision Notes**

The **major revisions to the initial submission** are summarized as follows：

- Fix the typos.
- Update the problem formulation in Sec 3 with an additional assumption.
- Add eward details in Sec 4.4.
- Replace the variance bar with confidence intervals in Figure 5.
- Update details of the generalisation experiment in Sec 6.2.
- Update the limitation and future work in Sec 7.
- Rename "Arrangement" task to "Rearrangement".

Besides, we have **demonstrated the additional analyses** on an [anonymous site](https://sites.google.com/view/neurips2022-paper2108-rebuttal) and **updated the supplementary** with the experiment details of the additional analyses (See Sec 5.6~5.9 in supplementary).

---

### Meta-Review · Area_Chair_8x2L · 2022-08-25

**Recommendation:** Accept
**Confidence:** Certain

**Metareview:**

After a strong rebuttal from the authors and an extensive discussion among the reviewers, I believe this work will be a valuable contribution to NeurIPS. I recommend it for acceptance and encourage the authors to address the reviewers comments for the camera-ready version of the paper, especially the point about the simplistic evaluation of the method - please consider a more realistic evaluation scenario.

**Award:**

No

---

### Decision · Program_Chairs · 2022-09-14

Accept